# Evaluation of Anti-*Candida albicans* Activity and Release of Ketoconazole in PMMA-G-PEG 4000 Films

**DOI:** 10.3390/ijms231810775

**Published:** 2022-09-15

**Authors:** Juliana Ribeiro Reynaldo, Kátia Monteiro Novack, Lucas Resende Dutra Sousa, Paula Melo de Abreu Vieira, Tatiane Roquete Amparo, Gustavo Henrique Bianco de Souza, Luiz Fernando Medeiros Teixeira, Ana Paula Moreira Barboza, Bernardo Ruegger Almeida Neves, Meiry Edivirges Alvarenga, Felipe Terra Martins, Viviane Martins Rebello dos Santos

**Affiliations:** 1Department of Chemistry, Federal University of Ouro Preto (UFOP), Ouro Preto 35400-000, MG, Brazil; 2Laboratory of Morphopathology, Center for Research in Biological Sciences, Federal University of Ouro Preto (UFOP), Morro do Cruzeiro Campus, Ouro Preto 35400-000, MG, Brazil; 3Postgraduate Program in Pharmaceutical Science, Federal University of Ouro Preto (UFOP), Morro do Cruzeiro Campus, Ouro Preto 35400-000, MG, Brazil; 4Physics Department, Federal University of Ouro Preto (UFOP), Morro do Cruzeiro Campus, Ouro Preto 35400-000, MG, Brazil; 5Department of Physics, Federal University of Minas Gerais (UFMG), Belo Horizonte 31270-901, MG, Brazil; 6Institute of Chemistry, Federal University of Goiás (UFG), Samambaia Campus, Goiânia 74690-900, GO, Brazil

**Keywords:** PMMA-g-PEG 4000, ketoconazole, *Candida albicans*

## Abstract

Modified release systems depend on the selection of an appropriate agent capable of controlling the release of the drug, sustaining the therapeutic action over time, and/or releasing the drug at the level of a particular tissue or target organ. Polyethylene glycol 4000 (PEG 4000) is commonly employed in drug release formulations while polymethyl methacrylate (PMMA) is non-toxic and has a good solubility in organic solvents. This study aimed at the incorporation of ketoconazole in PMMA-g-PEG 4000 and its derivatives, thus evaluating its release profile and anti-*Candida albicans* and cytotoxic activities. Ketoconazole was characterized and incorporated into the copolymers. The ketoconazole incorporated in the copolymer and its derivatives showed an immediate release profile. All copolymers with ketoconazole showed activity against *Candida albicans* and were non-toxic to human cells in the entire concentration tested.

## 1. Introduction

Different drug delivery systems have been projected that quantitatively deliver the drug to specific sites in the body and then trigger the release of the drug. The incorporation of the drug in the polymeric matrix is important for the release of the drug to occur correctly, and it is responsible for releasing the drug in specific sites in the body [1]. Drug delivery technologies modify the drug release profile, absorption, distribution, and elimination for the benefit of improving product efficacy, safety, as well as patient advances in polymer science, which have led to the development of several novel drug delivery systems. Polymers are becoming important due to their ability to favorably modulate drug delivery, and they have attracted significant academic research interest [2,3,4]. Polymeric release systems have introduced a new concept in drug administration to treat numerous diseases and maintain an adequate drug concentration in the blood or in target tissues for as long as possible and, with this, they are able to control the drug release rate [5,6,7,8]. During the past few decades, the occurrence of fungal infections has rapidly increased. Since fungal infections take a longer time to cure, the designing of drugs with extended persistence and modified release has gained attention [9]. These newer technological developments include chemical modifications, carrier-based delivery, and entrapment in polymeric matrices. These technical developments in drug delivery approaches improve human health [4]. Some pharmaceutical applications of polymers include: (a) taste masking; (b) as a binder in tablets for viscosity and flow controlling agent in liquids, suspensions, and emulsions; (c) film coating to disguise the unpleasant taste of the drug and to enhance drug stability; and (d) modified-released drug characteristics [10]. The chemical modification of a polymer aims to change its properties and behavior [11,12,13]. This change may influence the polymer-drug-organism interaction and promote better release of the active principle or prolong its action within the system. The copolymer of the present study combines characteristics of both polymers used to produce PMMA-g-PEG 4000. PMMA-g-PEG 4000 is thermoplastic, transparent, colorless, non-toxic, non-irritating, and has good solubility in organic solvents, which are characteristics derived from poly (methyl methacrylate) (PMMA) and polyethylene glycol (PEG). In addition, both polymers act as a vehicle for other drugs, increasing their half-life; moreover, the polymers do not possess drug-like activity, instead, they act to modulate the effects of actual drugs physically [14]. Some drugs do not fully arrive in their action site and interact with other tissues causing various adverse effects [15]. The study of polymers and their pharmacological use aims to reduce or even eliminate such effects that are undesirable during drug use [16]. The drug used in this study is ketoconazole, a broad-spectrum antifungal agent composed of a synthetic imidazole molecule, used for superficial or systemic infections. Despite the therapeutic efficacy already known, oral ketoconazole administration in high doses causes systemic liver ill effects, thus, the development of drug delivery systems with modified release is important. Medicated polymeric films have found a great application in topical therapy, since they are easily applied and avoid any trouble encountered in oral dosage forms [17]. The formulation of ketoconazole in polymeric films for topical application might present a novel drug delivery system [18]. Therefore, this study aimed at the incorporation of ketoconazole in PMMA-g-PEG 4000 and its derivatives (Figure 1), and to evaluate its release profile, as well as its antifungal and cytotoxic activities [19,20,21,22,23].

## 2. Results

### 2.1. AFM Analyses of PMMA-g-PEG and Derivatives with Ketoconazole Incorporated

Samples for AFM analysis were prepared by the spread coating method, using 0.7 mg/mL of dichloromethane solution on a mica substrate, and it was dried after 30 s by a nitrogen flux. In all cases, considerable changes on the film organization can be seen in Figure 2, produced by a modification of the polymer chain and the incorporation process. The AFM image also shows significant variations in sample roughness and in the film organization, probably due to the interaction between the drug and the copolymer.

### 2.2. TGA Analysis of PMMA-g-PEG 4000, PMMA-g-PEG 4000 Derivatives and with Ketoconazole Incorporated

The thermal stability of the obtained patches was tested using thermogravimetric analysis. The following parameters were determined: onset decomposition temperature and temperature corresponding to the weight loss of 50% (determined from TG curves). The weight of loss could be attributed to decomposition, dehydration, or volatization of the samples.

Thermogravimetric analysis (TGA) of the samples (Figure 3a,b) showed that the presence of ketoconazole decreases the thermal stability of PMMA-g-PEG 4000. It is valid to assume that the complex chemical structure of the drug interferes with the intermolecular molecule interactions between these polymeric chains, making their structures susceptible to thermal degradation. On the other hand, it was also observed that the thermal stability of acetylated and hydrolyzed derivatives increases with ketoconazole, which may be related to a greater affinity between their chains, favored by the inclusion of these chemical groups. Only the acetylated and hydrolyzed PMMA-g-PEG 4000, before incorporation, showed two levels of mass loss, the first stage between 100–300 °C and the second stage between 300–700 °C for the acetylated, and 50–300 °C and 300–450 °C for the hydrolyzate. These copolymers showed a weight loss content around 20% in the first stage and, in the second stage, 50% for the hydrolyzate and 80% for the acetylate. It was observed that ketoconazole shows two weight losses at 350–450 °C and 450–700 °C. It was also observed that only the hydrolyzed copolymer leaves 30% of residue after the end of heating. The addition of ketoconazole decreases the thermal stability of the samples, which can be proven by the increase in the stages of weight of mass in the samples. All samples showed initial weight loss at lower temperatures after drug addition, with contents ranging from approximately 3% to 10% mass loss in the first stage, in addition to showing two to three levels of mass loss until the end of the test. Only the ethylated and halogenated copolymers, incorporated, left around 10% of residue.

All PMMA-g-PEG 4000 with ketoconazole incorporated have two mass losses between 300–350 °C and 400–450 °C. It was observed that ketoconazole has two mass losses in 400 °C and 500 °C. The inclusion of the ketoconazole makes the copolymer less thermal resistant, showing three stages of mass loss. Although less thermal resistant, the incorporated copolymer chains have a lower overall mass loss content than the copolymer prior to incorporation of the drug.

### 2.3. DTA Analysis of PMMA-g-PEG 4000, PMMA-g-PEG 4000 Derivatives and with Ketoconazole Incorporated

In the DTA curves shown in Figure 4, it is possible to notice that almost all PMMA-g-PEG 4000 derivatives follow the PMMA-g-PEG 4000 pattern, with only the ethylated derivative showing a different behavior. It is possible to notice endothermic peaks that represent the melting of the raw material as well as exothermic peaks, probably related to a crystallization process. Based on the TGA results, it can be assumed that above 300 degrees, the highest number of peaks represents the subsequent thermal degradation of the sample. The DTA curves show the small influence of the drug addition, resulting generally in the displacement of the thermal transitions to higher temperatures, thus indicating a slight increase in the crystallinity of the samples, except for the PMMA-g-PEG4000.

### 2.4. Analysis XRD of PMMA-g-PEG 4000, PMMA-g-PEG 4000 Derivatives and with Ketoconazole Incorporated

The samples diffractograms show semi-crystalline material, therefore, they have an amorphous region below the crystalline peaks (Figure 5). It is possible to observe that the incorporation of ketoconazole increased the organization of PMMA-g-PEG 4000 hydrolized ketoconazole incorp and, consequently, their crystallinity. This increase may be related to the polar characteristic of the drug (Figure 6). Adding ketoconazole to PMMA-g-PEG 4000 induced greater amorphous nature (Figure 5 bottom curve and Figure 6 next to bottom curve), which is a behavior compatible with the results found in the analysis of TGA and DTA, and it indicated a decrease in the crystalline character of the sample after the incorporation of the drug. For the ethylated and acetylated derivatives, there were no significant changes observed in the diffractogram patterns with the addition of ketoconazole to the polymer carrier.

### 2.5. Analysis FTIR of PMMA-g-PEG 4000, PMMA-g-PEG 4000 Derivatives and with Ketoconazole Incorporated

Through the FTIR spectra, it is possible to confirm that there was incorporation of ketoconazole due to the presence of absorption bands in the IR spectra of the incorporated PMMA-g-PEG4000 and PMMA-g-PEG4000 derivatives. It is possible to observe the characteristic absorption bands of the ketoconazol structure at 3500 cm^−1^ related to the OH bond, two absorption bands at approximately 1600 and 1500 cm^−1^ related to the presence of aromatic C=C, the band related to the C=O bond of amide group at 1600 cm^−1^, and finally, the absorption band at 1000 cm^−1^ due to the presence of C–Cl bond as shown in Figure 7. The PMMA-g-PEG 4000 and pure PMMA-g-PEG 4000 derivatives showed the absorption band related to the C–OH bond at approximately 3500 cm^−1^ and C=O at 1600 cm^−1^, with larger bands and of stronger intensity in comparison with the IR spectra of PMMA-g-PEG4000 and PMMA-g-PEG4000 derivatives with ketoconazole incorporated, as shown in Figure 8. The stronger intermolecular interactions (drug–polymer), such as H-bonding and Van Der–Waals interactions, stabilize the system and decrease the vibration modes of the molecule.

### 2.6. In Vitro Release Study

After analyses, the release study was conducted (Figure 9). Release studies were performed at pH 7.2 to simulate the RPMI medium used in the cell viability assay. Precise amounts of polymeric films weighed on an analytical balance were transferred to volumetric flasks containing the buffer solution. Immediately after this process, the absorbance was evaluated by the Genesys 10S UV-Vis spectrophotometer. Pure ketoconazole was used as a control and showed maximum release (77%) in 90 min. The copolymer without chemical modifications (PMMA-g-PEG 4000 with ketoconazole incorporated) promoted a maximum release of 41% in 180 min, similar to the maximum concentrations promoted by the acetylated (42%) and halogenated (40%) copolymers, at the following times of 150 min and 105 min, respectively. The ethylated copolymer promoted a slower drug release reaching 34% at 225 min, suggesting that more ketoconazole could be released at unassessed times. The hydrolyzed copolymer promoted the release of 94% ketoconazole in 75 min, close to the maximum concentration that could be released, indicating a low interaction between the drug and the copolymer. Most curves show that ketoconazole is released in times close to 90 min, thus demonstrating an immediate release.

The lack of complete release of ketoconazole from the majority of polymer systems studied is directly correlated with drug–polymer interaction. There has to be an interaction, but it cannot be very strong as it is possible that the drug will not be released or undergo an incomplete release.

### 2.7. In Vitro Anti-C. albicans Evaluation

Following the incorporation and characterization of the copolymers incorporated, the biological activity was evaluated, and the corresponding minimum inhibitory concentration (MIC) values are shown in Table 1. All copolymers incorporated with ketoconazole showed activity against *C. albicans*. The copolymers (PMMA-g-PEG 4000, PMMA-g-PEG 4000 acetylated, PMMA-g-PEG 4000 hydrolyzed, PMMA-g-PEG 4000 ethylated, and PMMA-g-PEG 4000 halogenated) without ketoconazole (formulation negative control) were inactive against *C. Albicans*. This fungus is a member of the normal human microbiome but, using several virulence mechanisms, it can cause superficial infections of the skin (oral or vaginal candidiasis) and life-threatening systemic infections (bloodstream infections) [25].

Considering the proportion of the drug used, 5.2 times less than the positive control ketoconazole, the PMMA-g-PEG 4000 with ketoconazole incorporated, PMMA-g-PEG 4000 hydrolyzed with ketoconazole incorporated, PMMA-g-PEG 4000 ethylated with ketoconazole incorporated, and PMMA-g-PEG 4000 halogenated with ketoconazole incorporated were able to reduce the MIC. All these results of the PMMA-g-PEG 4000 films showed a significant difference (*p* < 0.05) when compared to ketoconazole not incorporated, according to the one-way analysis of variance (ANOVA) test followed by Dunnett’s test. Thus, the synthetized copolymers may represent therapeutic alternatives. These results deserve to be highlighted since the antifungal resistance of *C. albicans* is a growing health problem worldwide.

Ketoconazole is one of the effective antifungals used in the treatment of *C. albicans* infections, however, several resistant strains have been reported [26]. The synthetized copolymers may be therapeutic alternatives. The pharmaceutical preparations with PEG 4000 or PMMA have already shown an increase in the activity of antifungal drugs against *C. albicans*, but this is the first report of ketoconazole incorporated in PMMA-g-PEG 4000 copolymers [27,28,29].

### 2.8. Cell Viability

The oral administration of ketoconazole can lead to adverse effects such as systemic liver issues [30]. This toxicity may be related to the high lipophilicity, an important physico-chemical parameter that significantly contributes to the hepatotoxicity of drugs like ketoconazole [31]. Thus, the developed copolymers are an alternative to reducing the toxicity of this antifungal, since it reduces the lipophilicity of the drug due the polar terminal functional groups of PMMA-g-PEG 4000.

This low toxicity of the developed copolymers was indicated by the cytotoxicity assay performed by sulforhodamine B. All concentrations tested (62 to 1000 µg/L) were considered non-toxic for fibroblasts (Figure 10), according to ISO2009—10993-5, since the cell viability was higher than 70%. According to the two-way analysis of variance (ANOVA) test followed by post hoc Bonferroni test, there was no significant difference (*p* < 0.05) when comparing the results between the different PMMA-g-PEG 4000 films with ketoconazole incorporated.

The fibroblast is one of the most abundant cell types and it is present in tissues and organs such as the liver, skin, lungs, heart, kidneys, and eyes [32]. Therefore, the low toxicity at concentrations larger than the MIC (Table 1) indicate potential use of the PMMA-g-PEG 4000 derivatives for the treatment of infections caused by *C. albicans*.

## 3. Materials and Methods

### 3.1. Chemical Reagents

The reagents used for the films and incorporation with ketoconazole were sodium hydroxide (NaOH), acetic anhydride (CH_3_COOCOCH_3_), acetic acid (CH_3_COOH), hydrochloric acid (HCl), sodium bicarbonate (NaHCO_3_), sulfuric acid (H_2_SO_4_), commercial ketoconazole, and dichloromethane (CH_2_Cl_2_), purchased from Vetec. Poly (vinyl alcohol) (average mol wt 30,000–70,000), ethyl iodide (CH_2_CH_3_I), chloride sodium (NaCl), benzoyl peroxide, methyl methacrylate (MMA), and distilled water were purchased from Sigma Aldrich. The copolymers PMMA-g-PEG 4000 and derivatives were synthesized by our research group and published in scientific journals [19,20,21,22,23].

### 3.2. Incorporation of the Ketoconazole

The incorporation was realized in two phases. Organic phase: in a heater plate, a beaker with 6.00 mL of dichloromethane (CH_2_Cl_2_) was placed. When the solvent obtained a temperature of approximately 30 °C, 1.20 g of copolymer and 0.40 g of ketoconazole were added. This system was kept under stirring until complete dissolution. This solvent solubilized the PMMA-g-PEG 4000 and their derivatives at a temperature of 40 °C. Aqueous phase: a beaker of 100 mL containing 40.0 mL of water was warmed in a heating plate. When the temperature reached approximately 70 °C, 0.48 g of poly (vinyl alcohol) was added to the emulsified solution, stirring slowly until its complete dissolution. PVA was used as an emulsifier in the method. After the preparation of the two phases, the aqueous phase was poured into the organic phase. The mixture was subjected to 500 rpm of agitation for 4 h at a temperature of 35 °C. During this process, small portions (20 mL) of dichloromethane were added. After this time, the two phases were separated and the aqueous phase with PVA was discarded. Then, the organic phase (drug, polymer, and dichloromethane) was taken to the stove at 40 °C for 24 h in order to evaporate the dichloromethane. After this time, films (Figure 11) were obtained and submitted to the release process [20,21,22,23].


*After this time, the two phases were separated and the aqueous phase with PVA was discarded. Then, the phase organic (drug, polymer and dichloromethane).*


### 3.3. Characterizations

#### 3.3.1. Atomic Force Microscope (AFM)

The Atomic Force Microscope (AFM) characterization was conducted on a Bruker Multimode 8 SPM, using the intermittent contact imaging mode. Silicon cantilevers (NSC35/AlBS from Mikromasch), with spring constants of 5–15 Nm^−1^ and a tip radius of curvature ~10 nm, were used throughout the study for sample imaging. The analyses of in vitro release were done using UV/Vis spectroscopy equipment. Instruments Model FEMTO800Xi.

#### 3.3.2. Thermogravimetrics (TGA) and Thermal Differential (DTA) Analyses

The PMMA-g-PEG and its incorporated derivatives were evaluated by thermogravimetric equipment on a T.A. Instruments Shimadzu DTG 60/60 H Simultaneous DTA-TGA using a heating rate of 20 °C min^−1^, temperature range of 20–700 °C, and an aluminum crucible. The sample weights were 3 to 10 mg, and it was done in an inert atmosphere.

#### 3.3.3. X-ray Diffraction (XRD)

The X-ray diffraction (XRD) was made at room temperature and the samples were analyzed in the region of 5–70 °C (2Θ) and speed of 2 °C/I^−1^. The apparatus used was a Shimadzu diffractometer, Model XRD-6000, equipped with iron tube and graphite monochromator.

#### 3.3.4. Fourier-Transform Infrared Spectroscopy (FTIR)

The samples were analyzed using 100 mg of KBr and 4 mg of the complex. The spectra were obtained using the PerkinElmer Spectrum 400 equipment; 16 scans were performed in the region from 4000 to 220 cm^−1^ with a resolution of 2 cm^−1^. The technique used was transmittance.

### 3.4. In Vitro Release Study

In the process of incorporation with ketoconazole, the films of PMMA-g-PEG 4000 and their derivatives were obtained dried and submitted to the process of release. The concentration used for the release was 660.0 μg/mL for PMMA-g-PEG 4000 and their derivatives with ketoconazole and 127.0 μg/mL for pure ketoconazole, since the proportion of ketoconazole in the formulations is 19.23%. Buffer solution with pH 7.2 was used. Every 15 min, an aliquot of the solution was evaluated on the ultraviolet spectrophotometer (UV) at a wavelength of 269 nm, starting from 0 (zero) to at the most 4 h [22,23,24].

### 3.5. In Vitro Anti-C. albicans Evaluation

The in vitro anti-*C. albicans* property was evaluated against the yeast *C. albicans* ATCC 14408 by the broth microdilution method to establish the minimum inhibitory concentration (MIC) (CLSI, 2012). The yeast was cultivated in Saboraud-dextrose agar medium at 37 °C. The inoculum was prepared using the direct colony suspension method by means of a saline (0.9% NaCl) suspension of colonies selected from a 48-h agar plate, before each assay. The suspension was adjusted to reach turbidity equivalent to 0.5 of the McFarland standard scale (1 × 10^8^ CFU/mL). The inoculum was diluted in Roswell Park Memorial Institute (RPMI) 1640 in order to obtain a final assay with 2.5 × 10^3^ CFU/mL. The polymer solutions were made in RPMI1640 with 2% dimethylsulfoxide (DMSO) to obtain final concentrations from 1000.00 to 0.49 µg/mL. The 50 µL of the sample and 50 µL of inoculum were added in a 96-wells plate. In the negative control, 50 µL medium and 50 µL of inoculum were added. In order to verify the methodology efficacy, ketoconazole (250.00 to 0.12 µg/mL) was used as a positive control. Only 100 µL of the medium was added in the medium control. The plates were closed and incubated at 37 °C for 48 h. After the incubation period, 30 µL of triphenyl tetrazolium chloride (TTC, 0.25 mg/mL) were added and the plates were incubated for an additional 3 h. The MIC was established as the smallest concentration in which no yeast growth was detected (no visible color) [33].

### 3.6. Cell Viability

Human fibroblast MRC-5 cells, cultivated in RPMI 1640 medium (Sigma^®^, St. Louis, MO, USA), were distributed in 96-well microtiter plate using a density of 5 × 10^4^ cell/well and after, they were incubated at 37 °C with 5% of CO_2_ for 24 h. The cells were treated with the sample dissolved in RPMI, 2% dimethylsulfoxide (DMSO), at concentrations ranging from 1000.0 to 62.5 µg/mL. Before addition in the culture medium, the films were exposed to UV light at 254 nm for 20 min and no signs of microbial contamination were observed in the cultures during the tests. Cell viability was evaluated using the sulforhodamine B assay (SRB) [34]. After a 24 h incubation, the media was removed and the cells were fixed with cold 20% trichloroacetic acid for 1 h at 4 °C. The microtiter plate was washed with distilled water and dried. Thereafter, fixed cells were stained for 30 min with 0.1% SRB dissolved in 1% acetic acid. The plate was washed again with 1% acetic acid, and once again allowed to dry, while 200 µL of 10 mM Tris (hydroxymethyl) methyl aminomethane (TRIS buffer-pH 10.5) were added to stain solubilization at room temperature for about 30 min. The absorbance of the samples was read in the spectrophotometer (490 nm) and the results were expressed as the percentage of viable cells over untreated cells.

### 3.7. Statistical Analysis

The data were evaluated by one-way or two-way analysis of variance (ANOVA) using GraphPad Prism, version number 5.0. *p* values less than 0.05 were considered significant.

## 4. Conclusions

The current study showed the preparation of ketoconazole incorporated in PMMA-g-PEG 4000 and its derivatives. AFM analysis showed that important changes occurred after the drug incorporation process into the copolymers.

The TGA analyses indicate that the drug incorporation interferes with the initial temperature of degradation in the samples, making them less resistant to heat, however, the incorporated copolymer chains present a lower total mass loss content than the copolymer before incorporation of the drug, indicating an increase in the stability of the samples. DTA and XRD results demonstrated an increase in the crystallinity of the samples after drug incorporation, which may explain the increase in thermal stability observed in the TGA analyses.

The evaluation of ketoconazole release over 240 min showed that the copolymers released the most drug between 75 and 225 min. Furthermore, the formulations were not cytotoxic to human cells and an improvement in the antifungal activity of ketoconazole incorporated into the copolymers was observed with respect to the free antifungal, which was observed against *Candida albicans*.

## Figures and Tables

**Figure 1 ijms-23-10775-f001:**
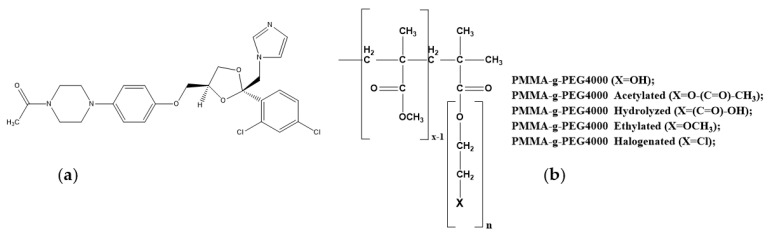
Chemical structure of the ketoconazole drug; (**a**) copolymer PMMA-g-PEG 4000 and derivatives; (**b**) [19,20,21,22,23].

**Figure 2 ijms-23-10775-f002:**
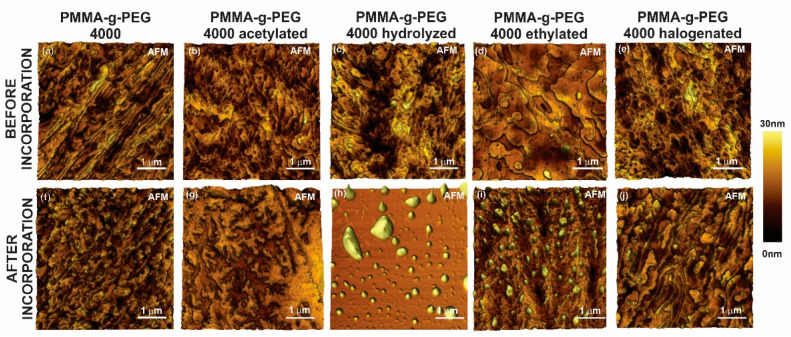
AFM images of PMMA-g-PEG 4000 and the derivatives thin films before (**a**–**e**) and after (**f**–**j**) the incorporation process.

**Figure 3 ijms-23-10775-f003:**
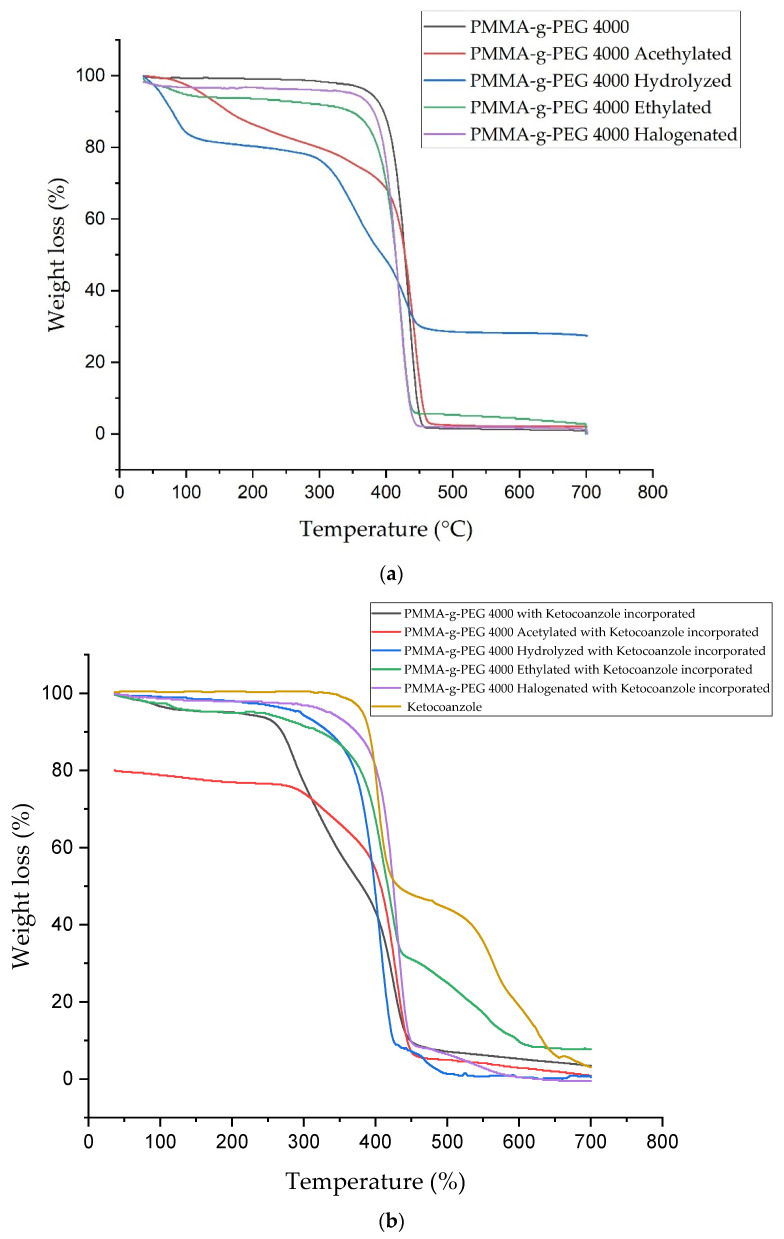
(**a**) TG curves of PMMA-g-PEG 4000 and PMMA-g-PEG 4000 derivatives; (**b**) TG curves of PMMA-g-PEG 4000 and PMMA-g-PEG 4000 derivatives with ketoconazole incorporated.

**Figure 4 ijms-23-10775-f004:**
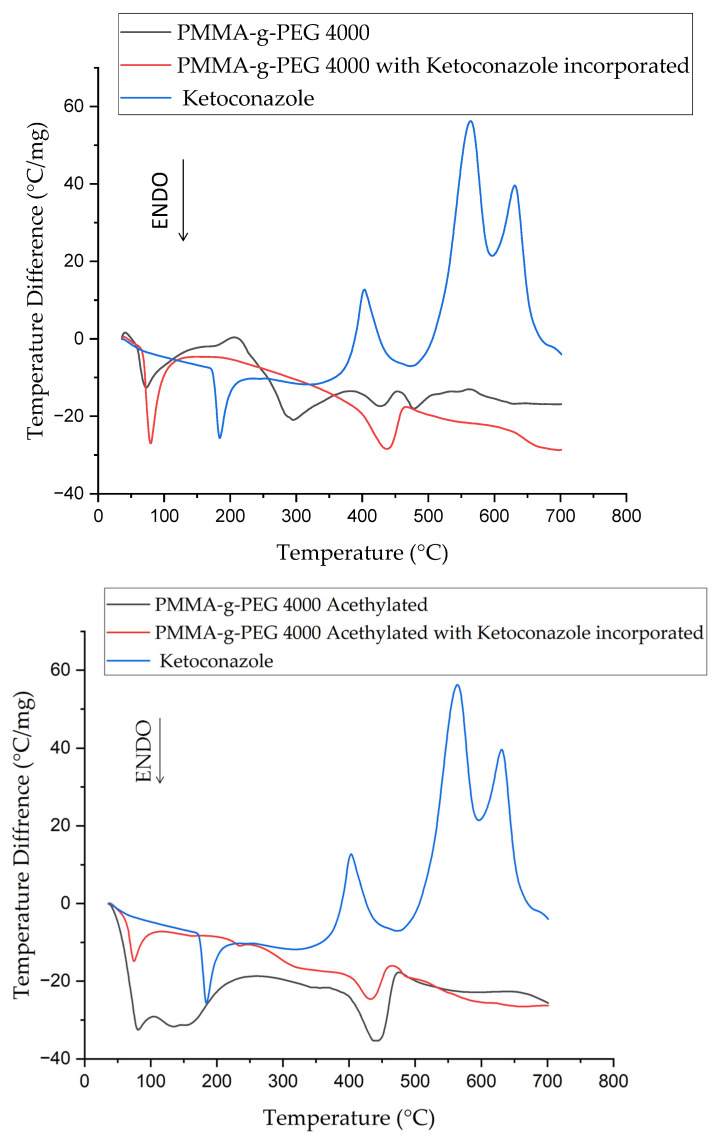
DTA curves of PMMA-g-PEG 4000, PMMA-g-PEG 4000 derivatives and with ketoconazole incorporated.

**Figure 5 ijms-23-10775-f005:**
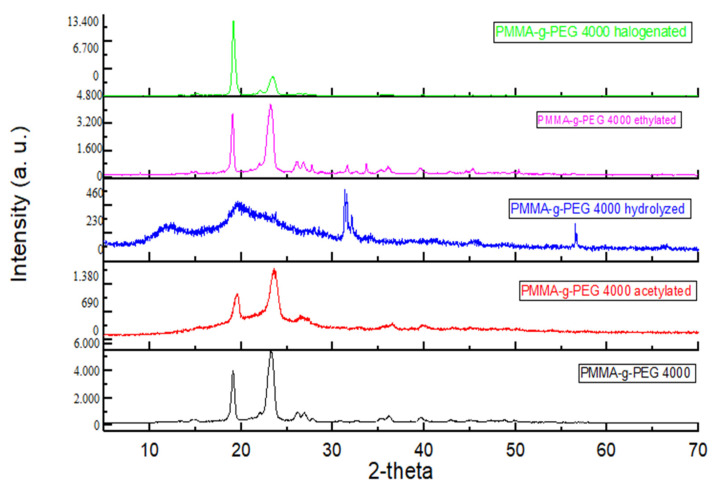
XRD of PMMA-g-PEG 4000 and PMMA-g-PEG 4000 derivatives.

**Figure 6 ijms-23-10775-f006:**
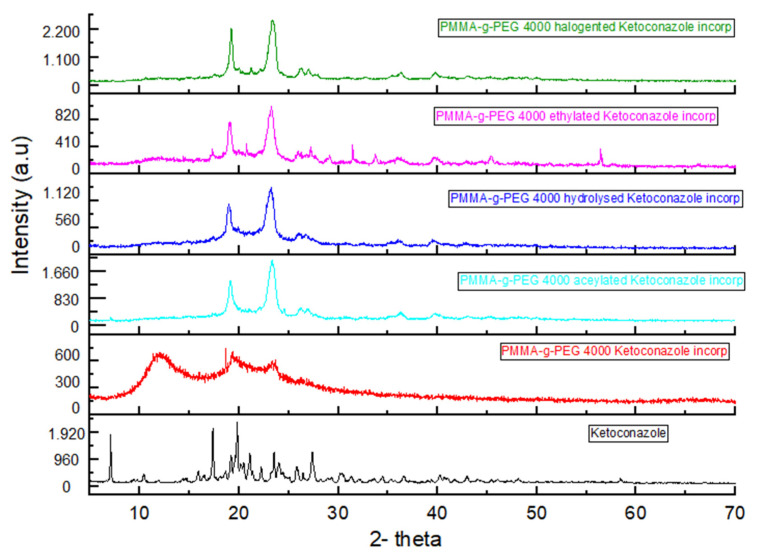
XRD of PMMA-g-PEG 4000 and PMMA-g-PEG 4000 derivatives with ketoconazole incorporated.

**Figure 7 ijms-23-10775-f007:**
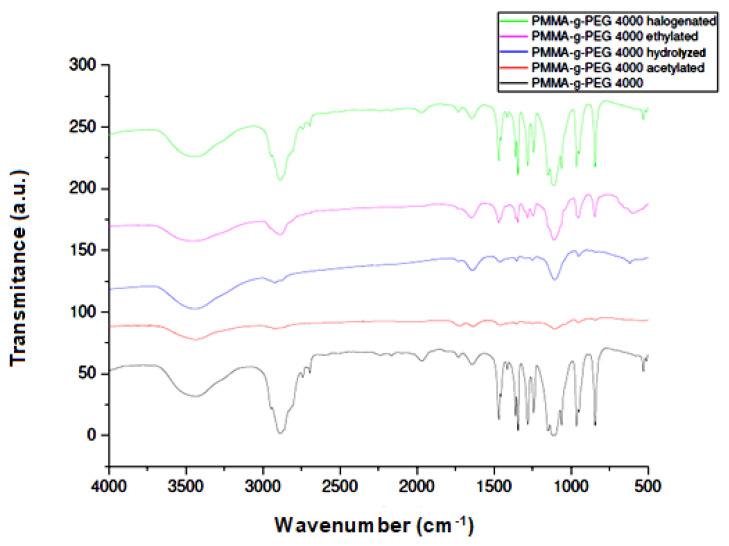
Infrared spectra of PMMA-g-PEG 400 and PMMA-g-PEG 4000 derivatives.

**Figure 8 ijms-23-10775-f008:**
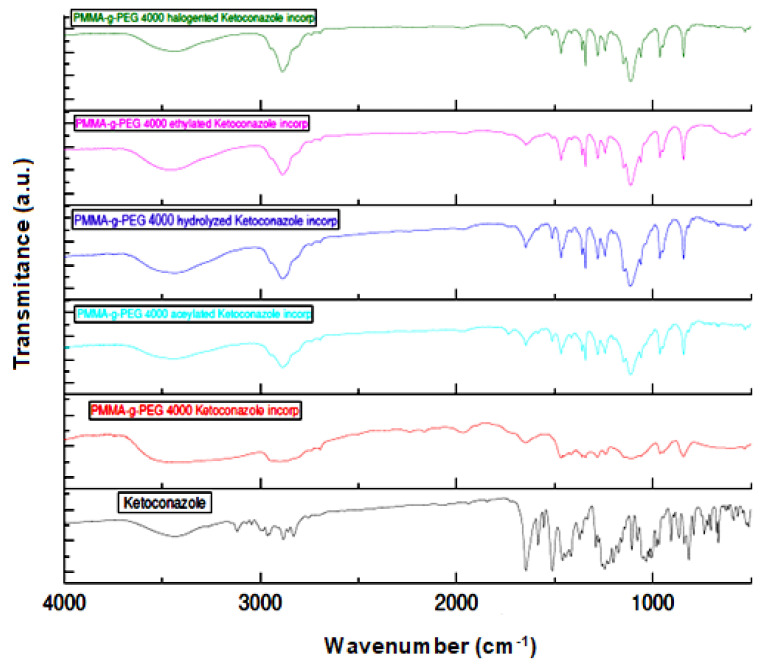
Infrared spectra of PMMA-g-PEG 400 and PMMA-g-PEG 400 derivatives with ketoconazole incorporated.

**Figure 9 ijms-23-10775-f009:**
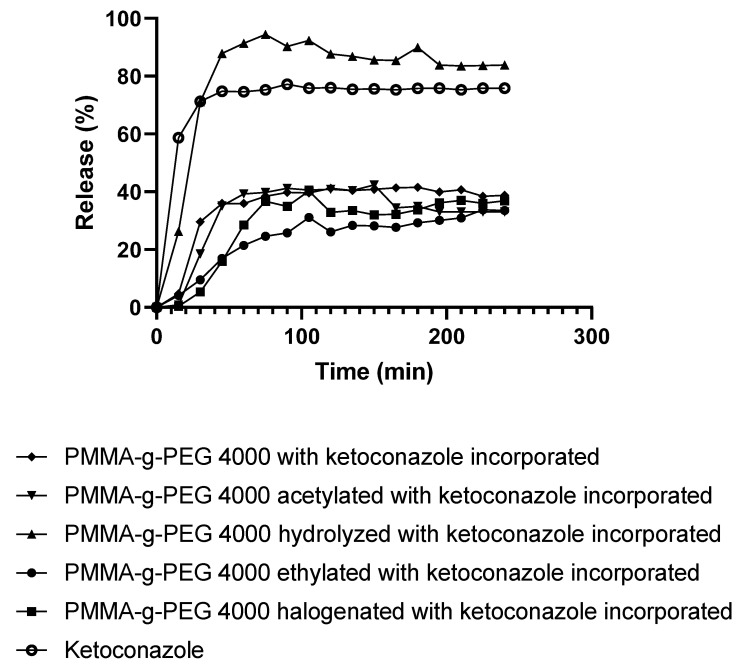
PMMA-g-PEG 4000 and derivatives with ketoconazole incorporated release curve at pH 7.2 for four h [24].

**Figure 10 ijms-23-10775-f010:**
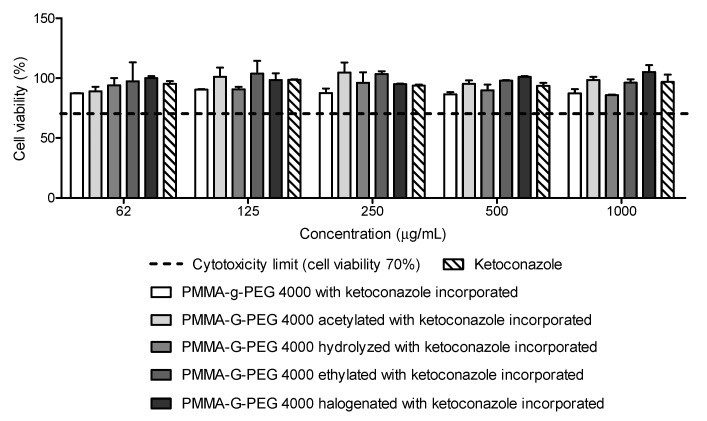
Cell viability of ketoconazole and PMMA-g-PEG 4000 derivatives.

**Figure 11 ijms-23-10775-f011:**
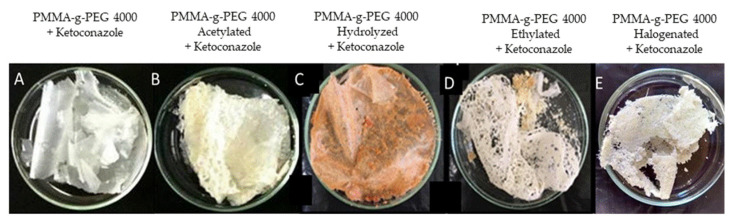
Images of PMMA-g-PEG 4000 and derivatives thin films after (**A**–**E**) the incorporation process.

**Table 1 ijms-23-10775-t001:** In vitro anti-*C. albicans* evaluation of PMMA-g-PEG 4000 and derivatives.

Sample	MIC (µg/mL)
Ketoconazole	125
PMMA-g-PEG 4000 with ketoconazole incorporated	500 * (96 ^#^)
PMMA-g-PEG 4000 acetylate with ketoconazole incorporated	1000 * (192 ^#^)
PMMA-g-PEG 4000 hydrolyzed with ketoconazole incorporated	500 * (96 ^#^)
PMMA-g-PEG 4000 ethylated with ketoconazole incorporated	500 * (96 ^#^)
PMMA-g-PEG 4000 halogenated with ketoconazole incorporated	500 * (96 ^#^)

* MIC of polymer with ketoconazole incorporated; ^#^ MIC according to the ketoconazole concentration in the formulation.

## Data Availability

Not applicable.

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
