# Peer review of "Evaluation of Anti-Candida albicans Activity and Release of Ketoconazole in PMMA-G-PEG 4000 Films"

_ijms, 2022, doi:10.3390/ijms231810775_

Round 1
Reviewer 1 Report
The authors have described the evaluation of the effectiveness of ketoconazole-loaded films based on novel PMMA-G-PEG 4000 copolymers against Candida albicans, such films claimed as offering controlled release properties to improve the potential clinical utility.
The concept is reasonable, and the authors have explored the copolymers in their earlier published work, so the manuscript is about investigation of the application of these polymers for a specific pharmaceutical application.
The manuscript is not suitable for publication in its present form. The authors will need to attend the the following issues to revise their manuscript to better suit it for further review and consideration for publication.
Page 1, Title: may need to spell out Candida albicans rather than C. albicans in the title to better support keyword searching.
Page 1, Introduction, lime 36-37: needs more explanation of the virtues of controlled release. To adapt the first sentence in a simple way, it might be more clearly described as a method of administering pharmaceuticals to achieve an enhanced therapeutic effect in humans or animals, relative to an immediate release profile, through modification of the time course of absorption or site-specific delivery.
Page 1, lines 40-44: authors may want to combine these sentences to talk of thee importance of polymers in drug delivery, having through ability to modulate drug delivery favorably have attracted significant academic research interest , then add further references here to complement the given 2, 3, and 4 which appear to be quite narrow in scope (all seem to be colonic delivery?). Additional references, covering scope including ant-fungal delivery will be useful?
Page 1 line 42: "career" should be "carrier"
Page 2 line 55-56: authors talk of the polymers having pharmacological activity. Surely, the polymers do not possess drug-like activity, they act to modulate the effects of actual drugs physically? Please clarify what the pharmacology you indicate here actually is or correct this section.
Page 2 line 83 - 96: how was the solvent evaporation controlled in each case to be sure the differences seen in the AFM work are not just due to the drying process?
Page 3 section 2.2: most if not all mass losses are above 300 degrees, so are any of these materials really thermally unstable, even those with ketoconazole?
Page 3 section 2.3 and Page 4, Figure 4: DTA curves above 300 degrees really show decomposition and may not be relevant to the discussion needed here, so could be replotted to more clearly show the important aspects of crystallization and melting transitions?
Page 3 section 2.4 and Figure 5 and Figure 6: the figures do not completely agree with the text in that ketoconazole does not seem to increase the organization of all polymeric chains. Adding ketoconazole to PMMA-G- PEG 4000 seems to induce greater amorphous nature (figure 5 bottom curve and Figure 6 next to bottom curve) and for the majority if other materials analyzed the XRPD patterns hardly change on adding ketoconazole to the polymer carrier. Please clarify how you arrived at eh conclusions given in section 2.4 about the effect of ketoconazole on polymer crystallinity.
Page 5, Section 2.6: unclear how samples were set up to undertake the controlled release studies, please add to experimental section how films were made and presented into the drug release test.
Page 6, figure 7: most, nearly all, of the curves show controlled release behavior,as all drug is released in 30- 90 minutes (i..e immediate release). Please clarify in the discussion.
Page 7 Table 1: are these MICs adjusted to the total amount of ketoconazole? You talk in the text about a reduction in the MIC but that is not reflected in the table. I suspect because the MICs in the table reflect the amount of drug/polymer use in the test, not the amount of drug?
Page 7,line 275 - Page 9 line 352: needs a lot more detail adding, OK that the synthesis of the polymers is referenced to authors prior work, but no detail on how controlled release drug films were made or how they were tested for controlled release. Section 3.2 describes how the drug/polymer powder is made ,but the paper title talks of controlled release films...they are not described clearly in the experimental section? Section 3.2.2 title says "termogravimetrics" which should be "thermogravimetrics". Give more details here of sample weights, nature of pan (open, closed, pinhole?). You say "usually" done in an inert atmosphere, under what criteria were non-inert atmospheres applied? Section 3.4, what apparatus was used for the drug release studies? A pharmacopoeial method or what? How were drug loaded films introduced into the test medium? How wee analytical samples filtered before UV analysis. How was the UV method qualified for use with these samples, for example to avoid interference by the polymers in the assessment of drug concentration? Was the concentration: absorption relationship linear over the range of concentrations tested? Section 3.5, how were films presented to the culture medium?
Author Response
Dear Editor and Reviewer 1
Please, find attached our revised "Evaluation of anti-C. albicans Activity and Controlled Release of Ketoconazole in PMMA-G-P
EG 4000 films " by Juliana Ribeiro Reynaldo, Kátia Monteiro Novack, Tatiane Roquete Amparo, Luiz Fernando Medeiros Teixeira, Lucas Resende Dutra Sousa , Paula Melo de Abreu Vieira, Ana Paula Moreira Barboza, Bernardo Ruegger Almeida Neves, Meiry Edivirges Alvarenga, Felipe Terras Martins and Viviane Martins Rebello dos Santos. We appreciate the comments and suggestions made about our manuscript. We have revised the manuscript. We believe these suggestions substantially improved this new version of the manuscript. All changes are highlighted in red, to ease their prompt identification.
Reviewer 2 Report
The manuscript presents the novel delivery system for ketoconazole to eliminate the limitation of the currently available oral dosage forms. The manuscript requires extensive revision before considering it for publication.
- The introduction section does not present the published reports on ketoconazole delivery, including films. Also, it does not explain the suggested form of delivery of the deleloped system.
- There are some standard methods for film production, e.g. film casting. What is the advantage of the used preparation method involving water phase and organic phase? It should be discussed in the manuscript. In typical emulsification process used to form micro- and nanoparticles, the PVA used as emulsifier is eliminated by several steps of washing. How was in this case? Does he PVA remain in the polymer film?
- Is the polymer film adequate name of the develoed DDS? The image of the developed film should be presented in the manuscript.
- What was the molar mass of the PVA?
- How was the film sterilized for cell culture study?
- What was the method of the evaluation of drug content in the polymer film?
- The statystical methods should be described.
- Figure 7 presents the drug released profile in µg/mL. Please, add the figure presenting the percentage release of drug. The bar errors should be added to present the SD. How many samples were used for drug release analysis?
- The results should be discussed in the detail in each section.
Author Response
Dear Editor and Reviewer 2
Please, find attached our revised "Evaluation of anti-C. albicans Activity and Controlled Release of Ketoconazole in PMMA-G-P
EG 4000 films " by Juliana Ribeiro Reynaldo, Kátia Monteiro Novack, Tatiane Roquete Amparo, Luiz Fernando Medeiros Teixeira, Lucas Resende Dutra Sousa , Paula Melo de Abreu Vieira, Ana Paula Moreira Barboza, Bernardo Ruegger Almeida Neves, Meiry Edivirges Alvarenga, Felipe Terras Martins and Viviane Martins Rebello dos Santos. We appreciate the comments and suggestions made about our manuscript. We have revised the manuscript. We believe these suggestions substantially improved this new version of the manuscript. All changes are highlighted in red, to ease their prompt identification.
Reviewer 3 Report
The authors worked on “Evaluation of anti-C. albicans Activity and Controlled Release of Ketoconazole in PMMA-G-PEG 4000 films”. The original research article is very well written and justified through suitable evaluation parameters and references. Though it contains sufficient results to be accepted for publication, but still modifications and certain studies are recommended to improve the quality of the manuscript.
Reviewer comments and suggestions;
1. The whole manuscript should be revised for grammatical and typographic mistakes.
2. Abbreviations should not be used in the abstract, please modify it.
3. Ketoconazole was mentioned with both small (k) and large letter (K) in the whole manuscript. Please mention it with large letter (K) throughout the manuscript.
4. The reagents used for the preparation of PMMA-G-PEG 4000 films should be explained with sufficient literature in the introduction section.
5. The problems should be addressed in detail in the introduction section while novelty should be explained at the last paragraph of the introduction for better understanding of the reader.
6. The resolution of Figure 3 should be enhanced. Also loss in weight of the formulation and its derivative should be mentioned in the text.
6. Both exothermic and endothermic peaks of the formulation and its derivative should be mentioned and explained while comparing with other published manuscripts.
7. In-vitro release study should be performed at pH 1.2 and 4.6 to know the release of the drug from the prepared films at various pH values.
8. It would be interesting if authors perform dissolution for any commercial available product of Ketoconazole and then compare with the formulation and its derivative to know the novelty of the prepared system.
9. To know the interaction between the formulation and drug, author should perform FTIR for drug and formulations.
Author Response
Comments and Suggestions for Authors
The authors worked on “Evaluation of anti-C. albicans Activity and Controlled Release of Ketoconazole in PMMA-G-PEG 4000 films”. The original research article is very well written and justified through suitable evaluation parameters and references. Though it contains sufficient results to be accepted for publication, but still modifications and certain studies are recommended to improve the quality of the manuscript.
Reviewer comments and suggestions;
- The whole manuscript should be revised for grammatical and typographic mistakes.
Authors. Done . Manuscript revised for ACE CONSULTING ENGLISH
- Abbreviations should not be used in the abstract, please modify it.
Authors: modified
- Ketoconazole was mentioned with both small (k) and large letter (K) in the whole manuscript. Please mention it with large letter (K) throughout the manuscript.
Authors: modified for (K)
- The reagents used for the preparation of PMMA-G-PEG 4000 films should be explained with sufficient literature in the introduction section.
Authors: done
- The problems should be addressed in detail in the introduction section while novelty should be explained at the last paragraph of the introduction for better understanding of the reader.
Authors: Medicated polymeric films have found a great application in topical therapy, since they are easily applied and avoid any trouble encountered in oral dosage forms [17]. Formulation of ketoconazole in polymeric films for topical application might present a novel drug delivery system [18].
References:
- Mohamed, M.S.; Eid, A.M.; Elgadir, M.; Mahdy M.A. Preparation
and release characteristics of Itracinazole poymeric films for topical
application. International Journal of Pharmacy & Pharmaceutical Sciences
2013, 5, 167-170. - Salama, M.; Mahdy, M. A.; Mohamed, A.; Mohamed, A.T.; Keleb, E. I.; Omar, A.A.; Elmarzugi, N. A. Formulation and Evaluation of Ketoconazole Polymeric Films for Topical Application. Journal of Applied Pharmaceutical Science 2015, 5 (05), 028-032.
- The resolution of Figure 3 should be enhanced. Also loss in weight of the formulation and its derivative should be mentioned in the text.
Authors: Done. Figure 3 enhanced.
Thermogravimetric analysis (TGA) of the samples (Figure 3 (a) and (b)) showed that the presence of Ketoconazole decreases the thermal stability of PMMA-g-PEG 4000. It is valid to assume that the complex chemical structure of the drug interferes with the intermolecular molecule interactions between these polymeric chains, making their structures susceptible to thermal degradation. On the other hand, it was also observed that the thermal stability of acetylated and hydrolyzed derivatives increases with Ketoconazole, which may be related to a greater affinity between their chains, favored by the inclusion of these chemical groups. Only the acetylated and hydrolyzed PMMA-g-PEG 4000, before incorporation, showed two levels of mass loss, the first stage between 100-300ºC and the second stage between 300-700ºC for the acetylated and 50-300ºC and 300 -450°C for the hydrolyzate. These copolymers showed a weight loss content around 20% in the first stage and, in the second stage, 50% for the hydrolyzate and 80% for the acetylate. It was observed that Ketoconazole shows two weight losses at 350-450ºC and 450-700ºC. It was also observed that only the hydrolyzed copolymer leaves 30% of residue after the end of heating. The addition of Ketoconazole decreases the thermal stability of the samples, which can be proven by the increase in the stages of weight of mass in the samples. All samples showed initial weight loss at lower temperatures after drug addition, with contents ranging from approximately 3% to 10% mass loss in the first stage, in addition to showing two to three levels of mass loss until the end of the test. Only the ethylated and halogenated copolymers, incorporated, left around 10% of residue.
- Both exothermic and endothermic peaks of the formulation and its derivative should be mentioned and explained while comparing with other published manuscripts.
Authors: Done. Figure 4 enhanced.
In the DTA curves shown in Figure 4, it is possible to notice that almost all PMMA-g-PEG 4000 derivatives follow the PMMA-g-PEG 4000 pattern, with only the ethylated derivative showing a different behavior. It is possible to notice endothermic peaks that represent the melting of the raw material and, also exothermic peaks, probably related to a crystallization process. Based on the TGA results, it can be assumed that above 300 degrees, the highest number of peaks represents the subsequent thermal degradation of the sample. The DTA curves show the small influence of the drug addition, resulting in a general way in the displacement of the thermal transitions to higher temperatures, thus indicating a slight increase in the crystallinity of the samples, except for the PMMA-g-PEG4000.
- In-vitro release study should be performed at pH 1.2 and 4.6 to know the release of the drug from the prepared films at various pH values.
Authors: We agree in-vitro release study should be performed at pH 1.2 and 4.6, but we made in-vitro release study it at pH 7.2 based on the following articles:
- Dos Santos, V.M.R.; NovacK, M. K.; De Sousa, D.V.M.; Oliveira, S.R.; Donnici, C.L. Síntese e Caracterização de novos copolímeros fosforilados. (2015). Polímeros: Ciência e Tecnologia, 25(suppl), 19-24.
- Dos Santos V.M.R.; Novack, K.M.; Silveira, B.M.; Marcondes, H.C. Preparation, Characterization, and Biological Activity Against Artemia Salina of News Copolymer PMMA-g-PEG Derivatives Incorporated With Fluconazole. Macromolecular Symposia 2018, 378(1), 1-8.
- Nascimento, L. G.; Lopes,S. A.; Teodolino, A. B. L.; Novack, K. M.; Barboza, A. P. M.; Nevas ,B. R. A.; Azevedo, M. L. S.; Sousa,L. R. D.; dos Santos, V. M. R. Novel PEG 4000 derivatives and its use in controlled release of drug indomethacin. Quimica Nova 2020, 43 (6), 685-691.
- Azevedo, M. L.S.; Silveira, B.M.; Novack, K.M.; Dos Santos, V.M.R. Study of controlled release of PMMA-g-PEG copolymer and derivatives incorporated with the indomethacin drug. Macromol.Symp. 2018, 381, 1800145-1800152.
- 24. Rapalli, V.k.; Banerjee, S.; Khan, S.; NathJha, P.; Gupta, G.; Dua, K.; Hasnain, M.D.S.; Nayak, A. K.; Dubey, S.K.; Singhvi, G. QbD-driven formulation development and evaluation of topical hydrogel containing ketoconazole loaded cubosomes. Materials Science & Engineering C 2021, 119, 1-15.
- It would be interesting if authors perform dissolution for any commercial available product of Ketoconazole and then compare with the formulation and its derivative to know the novelty of the prepared system.
Authors: The proposal is very interesting, we did made only on the study of controlled release of pure ketoconazole with copolymers and on biological assays. We did not have the formulation available in the laboratory. We only had pure ketoconazole
- To know the interaction between the formulation and drug, author should perform FTIR for drug and formulations. viviane
Authors: Done. FTIR for drug and polymers (Figure 7 and 8)
Through the FTIR spectra , it is possible to confirm that there was incorporation of Ketoconazole, due to the presence of absorption bands in the IR spectra of the incorporated PMMA-g-PEG4000 and PMMA-g-PEG4000 derivatives. It is possible to observe the characteristic absorption bands of the Ketoconazol structure at 3500 cm-1 related to OH bond, two absorption bands in approximately 1600 and 1500 cm-1 relating to the presence of aromatic C=C, at 1600 cm-1 the band of C=O bond of amide group, and finally, the absorption band at 1000 cm-1 due to the presence of C─Cl bond as shown in Figure 7. The PMMA-g-PEG and pure PMMA-g-PEG 4000 derivatives showed the absorption band related to the bond C─OH in approximately 3500 cm-1 and C=O in 1600 cm-1 with larger bands and of stronger intensity in comparison with the IR spectra of PMMA-g-PEG4000 and PMMA-g-PEG4000 derivatives with Ketoconazole incoporated, as showed in Figure 8.
Figura 7. Infrared spectra of PMMA-g-PEG 400 and PMMA-g-PEG 4000 derivatives
Figura 8. Infrared spectra of PMMA-g-PEG 400 and PMMA-g-PEG 400 derivatives with ketoconazole incorporated.
Round 2
Reviewer 1 Report
Authors have considered the feedback from this reviewer and used it to make modifications to the manuscript. The manuscript is not sufficiently improved to be suitable for publication as yet and the authors need to take the following on board in further revising the manuscript.
- It was pointed out in the previous round of review feedback that the in vitro release study indicates that drug is release from all samples tested, maybe excepting the ethylated co-polymer, within 90 minutes. so these materials do not provide for controlled release. Therefore the section 2.6 cannot really be titled " in vitro controlled release study, as there is limited or no control, and drug is released essentially in an immediate release fashion. Authors have indeed added a sentence that acknowledges this. But this section, section 3.4, the reference to control of release in the paper title, and the general discussion of control of drug release in the introduction, and reference to it in the abstract need to be rewritten in reflection of the results obtained. Yes, drug release from a polymer film is obtained but thee is little in the way of control. Rate may be slightly modified in the case of some materials, so there is indication that further work might enable more control of drug release.
- interesting results as to polymer crystallinity being modified by drug incorporation based on X-ray powder diffraction and the authors speculate the findings might be associated with drug-polymer interactions in the case of the hydrolysed polymer system (polymer becomes more crystalline through ketoconazole presence) or the unmodified (PMMA-g-PEG 4000) polymer (polymer becomes less crystalline). Discussion as to whether the FTIR absorption bands for ketoconazole shift in ways that would support the interactions speculated should be added to Section 2.5
Author Response
Editor and Review
Please, find attached our revised Manuscript entitled "Evaluation of anti-Candida. albicans Activity and Release of Ketoconazole in PMMA-G-PEG films" by Juliana Ribeiro Reynaldo, Kátia Monteiro Novack, Tatiane Roquete Amparo, Luiz Fernando Medeiros Teixeira, Lucas Resende Dutra Sousa , Paula Melo de Abreu Vieira, Ana Paula Moreira Barboza, Bernardo Ruegger Almeida Neves, Meiry Edivirges Alvarenga, Felipe Terras Martins and Viviane Martins Rebello dos Santos. We appreciate the comments and suggestions made about our manuscript. We have revised the manuscript. We believe these suggestions substantially improved this new version of the manuscript. All changes are highlighted in red, to ease their prompt identification. Some of our more detailed responses to the reviewers are commented here below in this letter
Thank you very much for all your kind attention.
Sincerely,
Dra. Viviane Martins Rebello dos Santos
Reviewer 2 Report
The revised version of the manuscript has been significantly improved. However, before publication, the two minor points shpuld be verified:
- Fig. 9 - the % cumulative release is described in the text, but the chart presents cumulative release ug/mL - please check
- The Authors explained that the PVA is eliminated from the films, however it is not obvious from the description of the films' preparation procedure, which says that after mixing the aqueous and organic phase the mixture was placed at 40 ºC for evaporation of the dichloromethane: "After the preparation of the two phases, the aqueous phase was poured into the organic phase. The mixture was subjected to 500 rpm of agitation, for 4 h at a temperature of 35 ºC. During this process, small portions (20 mL) of dichloromethane were added. Then, the solution mixture (drug, polymer and dichloromethane) was taken to the stove at 40 ºC for 24 h in order to evaporate the dichloromethane. After this time, films (Figure 11) were obtained and submitted to the controlled release process [20-23]."
Author Response

(The authors gave the same response as above.)

Round 3
Reviewer 1 Report
In section 2.6, discussion of in vitro release, there is little comment on the lack of complete release of ketoconazole from the majority of polymer systems studied. Please add a little more comment/discussion about the incomplete release of drug for most film formulations. Also if this can be referred back to in other sections, e.g. where the crystallinity of the drug in the film, or the interactions between drug and polymer suggested by vibrational spectroscopy is considered that would further improve the story given.
Author Response
Reviewer 1
Round 2
Comments and Suggestions for Authors
Authors have considered the feedback from this reviewer and used it to make modifications to the manuscript. The manuscript is not sufficiently improved to be suitable for publication as yet and the authors need to take the following on board in further revising the manuscript.
Reviewer 1:It was pointed out in the previous round of review feedback that the in vitro release study indicates that drug is release from all samples tested, maybe excepting the ethylated co-polymer, within 90 minutes. so these materials do not provide for controlled release. Therefore the section 2.6 cannot really be titled " in vitro controlled release study, as there is limited or no control, and drug is released essentially in an immediate release fashion. Authors have indeed added a sentence that acknowledges this. But this section, section 3.4, the reference to control of release in the paper title, and the general discussion of control of drug release in the introduction, and reference to it in the abstract need to be rewritten in reflection of the results obtained. Yes, drug release from a polymer film is obtained but thee is little in the way of control. Rate may be slightly modified in the case of some materials, so there is indication that further work might enable more control of drug release..
Authors: Modified.
2.6. In vitro Release Study
3.4. In vitro Release Study
Abstract: The copolymer and derivatives had a drug release profile
Introduction: Different drug delivery systems have been projected that deliver the drug quantitatively to the in specific sites in the body and then trigger the release of drug. The incorporation of drug to the polymeric matrix is importance for the release of the drug to occur correctly, and it is responsible for releasing the drug in specific sites in the body
Title: Evaluation of anti-Candida albicans Activity and Release of Ketoconazole in PMMA-G-PEG 4000 films
Reviewer 1: interesting results as to polymer crystallinity being modified by drug incorporation based on X-ray powder diffraction and the authors speculate the findings might be associated with drug-polymer interactions in the case of the hydrolysed polymer system (polymer becomes more crystalline through ketoconazole presence) or the unmodified (PMMA-g-PEG 4000) polymer (polymer becomes less crystalline). Discussion as to whether the FTIR absorption bands for ketoconazole shift in ways that would support the interactions speculated should be added to Section 2.5
Authors: Added in the text “Due to the stronger intermolecular interactions (drug-polymer ) such as H- bonding and VanDer – Waals interactions, those interactions stabilize the system and decrease the vibration modes of the molecule”.
Round 3
Reviewer 1: In section 2.6, discussion of in vitro release, there is little comment on the lack of complete release of ketoconazole from the majority of polymer systems studied. Please add a little more comment/discussion about the incomplete release of drug for most film formulations. Also if this can be referred back to in other sections, e.g. where the crystallinity of the drug in the film, or the interactions between drug and polymer suggested by vibrational spectroscopy is considered that would further improve the story given.
Authors: The lack of complete release of ketoconazole from the majority of polymer systems studied can is directly correlated with drug-polymer interaction. There has to be an interaction, but it cannot be very strong as it is possible that the drug will not be released or occur an incomplete release.